# Nanomaterials Respond to Lysosomal Function for Tumor Treatment

**DOI:** 10.3390/cells11213348

**Published:** 2022-10-24

**Authors:** Xuexia Tian, Anhua Shi, Hang Yin, Yutian Wang, Qiaoyan Liu, Wenling Chen, Junzi Wu

**Affiliations:** 1Department of Basic Medical, Yunnan University of Chinese Medicine, Kunming 650500, China; 2School of Clinical Medicine, Yunnan University of Chinese Medicine, Kunming 650500, China

**Keywords:** nanomaterials 1, lysosomal 2, tumor 3

## Abstract

The safety and efficacy of tumor treatment are difficult problems to address. Recently, lysosomes have become an important target for tumor treatment because of their special environment and function. Nanoparticles have unique physicochemical properties which have great advantages in tumor research. Therefore, in recent years, researchers have designed various types of nanoparticles to treat tumors based on lysosomal function and environment. In this review, we summarize and analyze different perspectives of tumor treatment, including direct destruction of lysosomes or lysosomal escape, drug delivery by nanoparticles, response to endogenous or exogenous stimuli, and the targeting of tumor cells or other cells. We describe the advantages and disadvantages of these approaches as well as the developmental prospects in this field. We hope to provide new ideas for better tumor treatment.

## 1. Introduction

Tumors are one of the leading causes of death worldwide. Many researchers are seeking ways to prevent and treat tumors in various fields. At present, conventional methods for tumor treatment include but are not limited to surgery, chemotherapy, radiotherapy and phototherapy, sonodynamic therapy, immunotherapy [1,2], gene therapy [3,4], traditional Chinese medicine therapy [5], interventional therapy and microwave therapy [6,7]. However, because of the complexity of tumors, they can easily metastasize and recur. In many cases, conventional treatments remain associated with adverse effects and limited efficacy. Clinical resistance to single treatments is commonly addressed by combined therapy, but combined therapy may increase the risk of toxicity and even threaten the patient’s life. Moreover, some categories of molecules such as hydrophobic drugs, radioisotopes, toxins and nucleic acids cannot achieve the desired therapeutic effect because of their instability or extensive off-target effects. However, it may be possible to overcome these limitations through the development of nanomedicine.

Recently, nanomedicine has shown great potential in tumor diagnosis and treatment. Nanoparticles are materials with a range of at least 1–100 nm [8,9]. They have unique physicochemical properties and can exist in various shapes. These properties allow them to be conjugated to different types of therapeutic molecules and adapted to various biological and clinical situations [10,11,12,13]. Thus, they can be designed with the appropriate size, shape and surface properties as nanocarriers for loading tumor drugs [14,15]. Nanocarriers can increase drug solubility and bioavailability and improve drug targeting in the tumor microenvironment. This can increase the local drug concentration in the tumor, ultimately enhancing the efficacy of the treatment combination. Many studies have shown that compared with free drugs, nanoscale drug delivery systems can target tumor cells more specifically with fewer side effects [16,17,18,19]. At present, numerous nanoparticle-based drug delivery systems have been approved for tumor treatment. Additional nano-drug delivery systems are undergoing clinical trials or preclinical evaluation. However, the transition from experimental to clinical application still faces many challenges, such as off-target toxicity and drug resistance. In systemic administration, only a small fraction of drug-loaded nanoparticles can be delivered to solid tumors. The remainder carry drugs to healthy tissues and organs through the blood circulation, resulting in off-target toxicity and side effects and causing damage to the body. Thus, there is an urgent need to explore new strategies to improve targeting and reduce drug toxicity. In recent years, with the in-depth study of lysosome function, many experts have come to believe that nanomaterials responding to lysosome function may become an important target for anti-tumor therapy.

The lysosome is a dynamically acidic (pH ≈ 4.8) vesicular compartment system responsible for the degradation of biological macromolecules [20]. Lysosomes are rich in hydrolytic enzymes. All received endogenous or exogenous macromolecules are degraded in the lysosome by more than 60 acid hydrolases and subsequently reused by the cell’s metabolic processes [21]. In addition to hydrolases, lysosomes also contain specific groups of intact membrane proteins and lysosome-associated proteins [22,23,24]. Lysosome-associated proteins can dynamically interact with the lysosome surface under certain conditions. Recent studies have found that lysosomes are involved in many other cellular processes in addition to being the “garbage disposal system” of the cell. These cellular processes include metabolism, gene regulation, immunity, plasma membrane repair and cell adhesion and migration [25,26,27]. Because of these functions, lysosomes have opened new perspectives in tumor research. First, in tumor cells, degradation pathways are deregulated, causing various alterations to the structure and function of lysosomal membranes. These changes ultimately make tumor cells more susceptible to stimulation by endogenous or exogenous factors that affect lysosomal membrane permeability (LMP) [28,29,30]. The increased permeability of the lysosomal membrane will release digestive enzymes into the cytoplasm. When lysosomal enzymes are released into the cytoplasm, lysosomal cell apoptosis (LCD) can be triggered in two ways. These two pathways are caspase-dependent and caspase-independent mechanisms [31,32]. Therefore, LCD is a strategy to kill the tumor cell. This strategy is still effective for tumor cells that are defective or resistant to the classical apoptosis procedures [29,33]. In addition, due to the acidic environment of lysosomes, the dissolution of lysosomes will release the contents to the cytoplasm, which will reduce the pH of the cytoplasm. And lysosomal dissolution will release drugs isolated in lysosomes, which can increase drug delivery. In conclusion, lysosomal lysis may enhance the therapeutic effect of tumors. Second, in gene therapy for tumors, the nucleic acid carried by nanocarriers is easily broken down after entering cells, which increases the many delivery challenges of therapeutic agents. Therefore, achieving lysosomal escape is also particularly important in tumor treatment. Overall, according to the function of the lysosome, different types of nanocarriers are designed, with corresponding treatment methods, providing possibilities for better tumor treatment.

There may be great utility in tumor treatment via lysosomes. This review addresses strategies utilizing different types of nanocarriers to treat tumors by destroying lysosomes or realising lysosomal escape. It will help readers better understand the application of nanocarriers in tumor treatment based on lysosomal function to provide appropriate therapeutic means for different types of tumors.

## 2. Destroying Tumor Cell Lysosomes and Killing Tumor Cells Directly

Lysosomes have become an attractive target in tumor therapy. At present, researchers are committed to using various means to destroy the function of lysosomes and directly lead to the death of tumor cells. Most of these studies are related to nano-drugs and nanocarriers directly killing tumor cells by destroying lysosomes in response to endogenous and exogenous stimuli.

### 2.1. Nanocarriers Respond to Exogenous Factors to Disrupt Lysosomes

#### Response to Magnetism

Zhang et al., developed a device capable of inducing and precisely controlling the rotation of magnetic nanoparticles around their axis [34]. This device is called a Dynamic Magnetic Field (DMF) Generator. The DMF generator generates a dynamic force field that induces rotational movements of superparamagnetic iron oxide nanoparticles (SPIONs). It also induces the rotational motion of SPIONs. Subsequently, researchers covalently conjugated SPIONs with antibodies targeting the lysosomal protein marker LAMP1 (LAMP1-SPION). Their findings suggest that long-range activation of the slow rotation of LAMP1-SPIONs significantly enhances the cellular internalization efficiency of the nanoparticles. LAMP1-SPIONs can bind to endogenous LAMP1. Consequently, LAMP1-SPIONs preferentially accumulate along the membrane in lysosomes of rat insulinoma tumor cells. The subsequent remote activation of the DMF could cause the rotating sheer force of LAMP1-SPIONs. This mechanical shear force mechanically tears the lysosomal membrane. This result could cause the exosmosis of lysosome contents into the cytoplasm and a decrease in intracellular pH, which would eventually lead to apoptosis. DMF treatment of lysosome-targeted nanoparticles provides a non-invasive tool for remote induction of apoptosis and can serve as an essential platform technology for tumor therapy. Moreover, Domenech et al. conjugated iron oxide (IO) magnetic nanoparticles (MNPs) to the protein epidermal growth factor receptor (EGFR) [35], which recognizes receptors that are overexpressed in cell membranes. EGFR is the expression product of proto-oncogene c-erbB1 and is a member of the epidermal growth factor receptor family, which is highly expressed in many tumors. The results showed that IOMNPs binding to EGFR can specifically recognise tumor cells with high expression of EGFR. The targeted nanoparticles were then internalized into the lysosome. Researchers can observe EGFR-binding IOMNPs in cell membranes and lysosomes. When an alternating magnetic field (AMF) is applied, magnetic nanoparticles vibrate, and when they vibrate, they generate heat locally. It causes LMP, which releases lysosome contents into the cytoplasm. It has been proved that MNPs targeting EGFR under the action of AMF can selectively induce LMP in tumor cells that overexpress EGFR, and finally kill tumor cells. Studies have demonstrated that EGFR-targeting MNPs under the action of AMF can selectively induce LMP in EGFR-overexpressing tumor cells and kill tumor cells [35].

Triple-negative breast cancer (TNBC) is one of the deadliest subtypes of breast cancer [36,37]. TNBC refers to a special type of breast tumor that is negative for the estrogen receptor (ER), progesterone receptor, and human EGFR (HER2). Targeting tumour cells of TNBC is a challenge due to the lack of these receptors [38,39,40]. Poly (ADP ribose) polymerase (PARP) inhibitors are a class of medically useful agents capable of affecting the self-replicative manner of tumor cells. They can largely inhibit the repair of DNA breaks in tumour cells and then promote tumor cells to undergo apoptosis. PARP inhibitors are a novel strategy in the treatment of TNBC [41]. Olaparib (Olb) is the first PARP inhibitor approved for clinical use [42]. However, its clinical outcome is highly controversial, and the results show insignificant anticancer effects of Olb in wild-type BRCA tumors [43]. This might be due to poor drug targeting and low bioavailability [44,45,46]. Therefore, developing an effective oral drug delivery platform to improve drug bioavailability and achieve specific targeting is a promising strategy for TNBC treatment. Multifunctional nanomaterials are used to optimize drug delivery and tumor targeting because of their unique properties. Zhang et al., developed PEI-PLGA-based "cell-targeted disruptive" multifunctional polymer nanoparticles (named HA-Olb-PPMNPs) (Figure 1) [47]. The nanoparticles were loaded with Olb and superparamagnetic IO nanoparticles (Fe^3^O_4_ NPS). CD44 molecules are expressed to varying degrees on the surface of many types of tumor cells. Hyaluronic acid (HA) has a high affinity with the CD44 receptor [48,49]. The findings revealed that HA-Olb-PPMNPs could specifically recognize tumor cells. Under a rotating magnetic field (RMF), HA-Olb-PPMNPs generate mechanical forces through incomplete rotation. This mechanical force shears and disrupts the cell membrane when it internalizes HA-Olb-PPMNPs. This process, called magnetic cytolysis, is the “first hit”. This mechanical force, in turn, damages lysosomes. Lysosomes, when disrupted, activate the lysosome mitochondria pathway. Ultimately, apoptosis will ensue. This process, known as ferroptosis, is the “second hit” [47]. Therefore, in the presence of RMF, mechanical force and Olb exert dual antitumor effects to achieve synergistic treatment. This study proposes a new multiple treatment concept for TNBC. It also provides evidence for the induction of new antitumor therapeutic effects by the magnetic nanomedicine system.

### 2.2. Nanocarriers Respond to Endogenous Factors to Disrupt Lysosomes

#### Response to pH

Borkowska et al., investigated a type of nanoparticles covered with different ratios of positively and negatively charged ligands, termed mixed-charge nanoparticles [50]. This hybrid nanoparticle uses gold nanoparticles with d = 5.3 ± 0.7 nm combined with positively charged N, N, N-trimethyl (11-mercaptoundecyl) ammonium chloride (TMA) and a negatively charged 11-mer-captoundecanoic acid (MUA) ligand for functionalization. Depending on the balance of “+” and “−”, the mixed-charge nanoparticles can precipitate or even crystallize in vitro at different pH values. Additionally, there is a difference in pH between cancerous tissue/cells and normal tissues/cells. According to this property, the pH-dependent aggregation of [+/−] NPs could be useful for selective targeting of tumor cells or their compartments. These findings suggest that mixed nanoparticles first gather on the surface of tumor cells. Next, these hybrid nanoparticles are internalized into ~50–100 nm nanoparticle clusters by endocytosis. These NP clusters gradually accumulate in multivesicular endosomes and are then transported to the lysosomes. In the lysosomes, mixed nanoparticle clusters are pH-dependent and assemble into ordered nanoparticle super crystals. Nanoparticle super crystals induce osmotic flow and lysosomal swelling. The lysosomal membrane gradually loses its integrity, eventually leading to cell death. In contrast, in normal cells, the aggregation of mixed nanoparticles is limited. Finally, they are expelled from the cell by exocytosis, with less damage to normal cells. This demonstrates selective lysosome targeting in which [+/−] NPs gradually disrupt the integrity of lysosomal membranes, ultimately triggering lysosome-dependent cell death selectively in cancerous cells.

Yang et al., proposed a new design of nano prodrugs with a “degradation-mediated self-toxification” strategy [51]. It can use the nanoparticles’ own degraded fragments as low-toxicity drug precursors to realize the intracellular synthesis of antineoplastic drugs. Researchers first screened metal-complexed dicyclohexylphosphine (DCP) organosilanes from a variety of ligands and coupled them to PD (OH)_2_ nanodot-confined hollow silica nanospheres (PD-HSN). The results showed that the constructed nano-drug was degraded in acidic lysosomes and the environment. Nano-drugs produce less-toxic fragments of Pd^2+^ and DCP silicates after degradation [51]. These fragments form DCP/PD complexes in situ and these DCP/Pd complexes bind to DNA strands and are cytotoxic, eventually leading to cell damage. Additionally, nano-drug degradation destroys the acidic environment of tumor cell lysosomes, leading to LMP. These self-toxifying nano-drugs can kill tumor cells and enhance anticancer efficacy in the 4T1, MCF-7, and CT-26 tumor cell lines.

Photodynamic therapy (PDT), which involves a combination of photosensitizers, light and molecular oxygen (^3^O_2_), is an emerging treatment modality for various tumors [52]. Temporal and spatial control of singlet oxygen (^1^O_2_) release is a major challenge in photodynamic therapy for cancer treatment [53]. For this reason, Tian et al. designed and synthesised selenium–rubyrin (NMe_2_Se_4_N_2_)-loaded nanoparticles functionalized with folate (FA) (Figure 2) [54]. This nanoparticle can act as an acidic pH-activatable targeted photosensitizer. Folate receptors are overexpressed on the cell membrane of most tumor cells [55]. The FA on the nanoparticles can specifically bind to the FA receptor and target tumor cells. The nanoparticles are then endocytosed by tumor cells. Nanoparticles will not be activated under physiological pH conditions. Due to the acidic lysosome environment, nanoparticles are activated when they reach lysosomes. Nanoparticles respond to the characteristics of lysosomal acidic environment to realize the pH controllable activation of NMe_2_Se_4_N_2_. When activated, nanoparticles produce ^1^O_2_ that destroys lysosomes and eventually triggers tumor cell death in lysosome-related pathways. In addition, the controllable activation of pH makes nanoparticles targeted to kill tumor cells, while reducing damage to normal cells, which improves the safety of nanoparticles in the human body [54]. This greatly reduces side effects. Tumor elimination was observed after near-infrared irradiation by intravenous injection of FA-NMe_2_Se_4_N_2_ nanoparticles in tumor-bearing mice. This experiment provides a new strategy for improving the antitumor effect of PDT. It also provides a new idea for the development of photosensitizers.

Non-small cell lung cancer (NSCLC) is one of the most frequently diagnosed cancers and the leading cause of cancer-related deaths worldwide [56]. Moreover, traditional chemotherapy treatments are not effective in treating NSCLCs. Recently, LMP could be induced in tumors [57]. Therefore, lysosomes have attracted a lot of attention as novel therapeutic targets. However, lysosomal inhibitors suffer from two disadvantages: poor targeting and lower bioavailability. Therefore, overcoming these two drawbacks is the key to achieving tumor treatment by modulating lysosomal function. Baehr et al. studied a deformable nanomaterial composed of polypeptide amphiphilic molecules called cell-penetrating transformable peptide nanoparticles (CPTNPs) [56]. CPTNPs contain a poly-d-arg motif (8-mer), an all D-amino acid containing β sheet-forming motif (kffvlk), and BIS pyrene. Thus, this nanomaterial can achieve colocalization with lysosomes and respond to changes in pH. CPTNPs enter tumor cells via reticulon-mediated endocytosis and reach the lysosomes. Due to the low pH of lysosomes, CPTNPs change into nanofibers in response to the change in pH, resulting in the release of LMPs. The lysosomal contents are released concomitantly sequestered with cisplatin, which enhances the effect of cisplatin in tumor cells. This therapeutic strategy produced a synergistic antitumor effect. This study, to the best of our knowledge, is the first to explore a transformable peptide nanomaterial responsive to lysosomal function, holding great promise for the study of tumors.

With good biocompatibility, biodegradability and photothermal conversion, polydopamine (PDA) has potential in tumor diagnosis and targeted drug delivery [58,59]. However, PDA still faces many challenges, such as its nanobioactivity in tumor cells. To investigate the mechanism of action and nanobioactivity of PDA-encapsulated nanoparticles in tumor cells, Ding et al. designed nanoparticulates with PDA-coated mesoporous silica nanoparticles (MSNs), termed PDNPs [60]. The researchers then investigated the transport mechanism of PDNPs by labelling them with different markers. PDNPs were internalized through three specific pathways in the HeLa cell line model. These three pathways include Caveolae-dependent and Arf6-dependent endocytosis and Rab34-mediated micropinocytosis. The autophagy-mediated accumulation of PDNPs in the lysosome and the formed PDA shell shed in the lysosome were experimentally observed. The study revealed that almost 40% of the NPs were transported out of the cells by exocytosis. Based on these results, Ding et al. proposed a novel combinatorial tumor therapy strategy using drug-loaded MSNs-PDA involving: (1) organelle-targeted drug release in lysosomes to generate lysosomal damage by exploiting natural intracellular machinery-controlled PDA shedding, and (2) blockage of proven exocytosis pathways to enhance therapeutic efficacy.

### 2.3. Nanocarriers Destroy Lysosomes by Other Means

Pyroptosis is a lytic form of programmed cell death [61]. In recent years, it has attracted attention because of its relationship with the pathogenesis of tumors [62,63]. Pyroptosis is triggered by the inflammasome [64]. Inflammasomes are intracellular multiprotein complexes. Among them, the nucleotide-binding oligomerization domain-like receptor protein 3 (NLRP3) inflammasome is a well-documented regulator of pyroptosis [64]. The activation of NLRP3 inflammasome promotes the self-cleavage of procaspase-1 into active caspase-1. Caspase-1 can cause gasdermin-D (GSDMD) cleavage, leading to pore formation on the cell membrane [65]. Water influx into the cell membrane then leads to cell-swelling induced plasma membrane rupture. The subsequent release of cellular contents eventually triggers pyroptosis. The activation of NLRP3 can be caused by interconnected cellular pathways, such as intracellular reactive oxygen species (ROS) production and lysosome disruption. To this end, Nadeem et al. designed a virus-spike tumor-activatable pyroptotic agent (VTPA) that has a virus-like, spike-like morphology and can generate ROS [66]. The researchers used organosilica to coat IO nanoparticles and spiky manganese dioxide protrusions. Following systemic administration, VTPA accumulates in tumors and promotes lysosomal rupture of tumor cells. Subsequently, VTPA is degraded by overexpressed glutathione (GSH) in tumor cells. The degraded VTPA releases Mn ions and iron oxide nanoparticles (IONPs). IONPs synergistically activate the NLRP3 inflammasome (Figure 3). The experimental results showed that VTPA treatment could activate NLRP3 inflammasome and release lactate dehydrogenase in tumor cells, eventually leading to specific pyroptosis. The structure-dependent and tumor intracellular GSH-activatable pyroptotic agents represent the first demonstration of cancer-specific pyroptosis in vivo. They also provide a new paradigm for the development of next-generation cancer-specific pyroptotic nanomedicines.

### 2.4. Low-Toxicity Nanomedicine Directly Destroys Lysosomes

In the past few decades, the continuous development of nanomedicine has made many contributions to tumor treatment. Researchers are developing carrier-assisted and carrier-free nanomedicines to improve drug intrinsic kinetics and safety [67,68,69,70,71]. However, these nanotherapeutics still have some limitations. For example, due to the multicomponent nature of nanotherapy, its complexity and toxicity seriously hinder the clinical transformation of many nanopreparations [72]. Some drugs that were used in nano-carrier delivery studies are no longer suitable for clinical anti-tumor therapy [73,74,75]. Additionally, due to the complexity of drug structures, they are difficult to modify. These limitations hinder the development of nano-drugs, but these problems are likely to be solved with pharmaceutical chemistry [76]. Nanomedicine mainly focuses on nano-drugs and nanocarriers. Pharmaceutical chemistry is committed to finding safe and effective drugs [76,77]. Drug discovery technology has developed a large number of lead drugs and drug candidates. However, it remains challenging in terms of pharmacokinetics, metabolism and toxicology [78,79]. We can find solutions to these challenges through the field of nanomedicine. Therefore, Ma et al. leveraged the relationship between these two disciplines to combine nanotechnology with initial drug design to develop a one-component new-chemical-entity nanomedicine (ONN) strategy [80]. This strategy not only follows the traditional drug design strategy but also the principle of molecular self-assembly. Drugs designed with this strategy have advantages in terms of drug discovery and delivery. They designed a series of lipophilic cationic bisaminoquiniline (BAQ) derivatives by pharmacophore hybridisation and molecular self-assembly [81,82]. Because BAQ12 and BAQ13 have the potential to be therapeutic agents and are self-assembled components, they were chosen as ONNs. Studies have found that these BAQ ONNs can destroy lysosomes, lead to lysosome dysfunction and inhibit autophagy (Figure 4). BAQ ONNs showed a clear antitumor effect in vitro. Additionally, as nano-drugs, BAQ ONNs are adept at self-delivery [80]. From a drug discovery and drug delivery perspective, these advantages enable the remarkable anticancer activity of BAQ ONNs as a single agent in an in vivo gastrointestinal cancer model.

## 3. Achieving Lysosomal Escape of Tumor Cells

Nanoparticles can extend the half-life of drugs or nucleic acids, and they preferentially accumulate in tumors. They enhance cellular uptake and allow for stimulus-responsive payload release [83,84,85]. However, extracellular sources are easily degraded in the lysosomes after being endocytosed by tumor cells. Nanoparticles deliver drugs or nucleic acids into cancer cells to penetrate the cell membrane, escape the lysosome and enter the nucleus. Therefore, achieving lysosomal escape is a critical step for nanotechnology to fully play a role in tumor treatment.

### 3.1. Nanocarriers Respond to Endogenous Factors Triggering Lysosomal Escape in Tumor Cells

#### Response to pH

The RNA interference (RNAi) of small interfering RNA (siRNA) can target almost any gene [86,87,88,89]. It has the potential to treat a variety of diseases, including tumors [90]. However, few RNAi therapies have undergone phase II/III clinical trials. This is due to the poor stability of naked RNA in blood. It does not enter cells and is poorly stable in the endo/lysosome. Nanotechnology holds great potential for RNAi therapy. Nanoparticles could extend RNA half-life in blood and preferentially accumulate in tumors. Nanoparticles can enhance cellular uptake and release payloads in response to certain stimuli. However, the RNAs released from nanoparticles can be easily degraded in endosomes/lysosomes after cellular uptake by endocytosis [91]. Therefore, achieving endo/lysosomal escape is critical for the efficient release of siRNAs into the cytoplasm. In order to improve the bioavailability and endo/lysosome escape of siRNA, Xu et al. designed “nanobomb” nanoparticles activated by low pH value to deliver POLR2A siRNA (siPol2) for the treatment of TNBC [92]. TNBC is the only breast cancer without an approved targeted therapy [93,94]. TP53 is the most frequently mutated or deleted gene in TNBC [95,96]. Through in silico analysis, researchers identified POLR2A in the vicinity of the TP53-neighboring region as a collateral vulnerability target in TNBC tumors. This suggests that inhibiting it by siRNA might be a viable approach for TNBC-targeted treatment [92]. When the pH in the endo/lysosome reduces, CO_2_ could be produced from the nanoparticles to produce the nanobomb effect. This effect triggers endo/lysosomal escape to enhance cytoplasmic siRNA delivery. The results showed that the inhibition of POLR2A with siPol2-loaded nanoparticles (siPol2@NPs) resulted in slowed growth of tumors characterised by hemizygous POLR2A loss.

### 3.2. Nanocarriers Respond to Exogenous Factors Triggering Lysosomal Escape in Tumor Cells

#### Response to Light

Xue et al., prepared a new photosensitive monofunctional Pt complex, Pt-BDPA, with the BODIPY chromophore bearing a Pt chelator [97]. This complex can bind to DNA. It can also display emissions at about 578 nm. Its singlet oxygen quantum yield is 0.133. Confocal imaging shows that this complex is sequestered in the lysosome by endocytosis in the dark and cannot enter the nucleus. After light-induced generation of ROS, the complex can undergo lysosomal escape into the nucleus upon light irradiation [97]. Meanwhile, light-induced ROS can reduce intracellular GSH levels to stabilize Pt-BDPA in the cytosol. This increases the nuclear DNA accessibility of this platinum complex and enhances its antitumor activity. Studies show that in the absence of radiation, Pt-BDPA is sequestered in lysosomes. They currently show poor cytotoxicity to normal cells. The study provides the first example of a Pt complex capable of light-activated lysosome escape. It offers the potential for precision therapy via selective light-activated lysosome escape of lysosome-sequestered Pt complexes in tumor tissues.

## 4. Achieve Lysosomal Escape of Immune Cells

Tumor immunotherapy prompts immune cells to kill tumor cells without harming normal tissue. Immunotherapy could activate immune cells. Activated immune cells systematically target primary tumor cells and secondary metastases [98]. One million tumor cells must be targeted in conventional treatment. However, immunotherapy only requires the activation of a few thousand immune cells. Therefore, immunotherapy has better efficacy and fewer toxic side effects. Immunotherapy can also avoid drug resistance to a certain extent. At present, immunotherapy through lysosomal escape has become an effective tool for tumor treatment.

### 4.1. Nanocarrier Response to Endogenous Factors Triggering Immune Cell Lysosomal Escape

#### Response to pH

T cells play a vital role in tumor monitoring and killing and are called “special forces” in the human body. Immunotherapy, which promotes T cells to kill tumor cells, has become a new therapeutic pillar in oncology. However, T-cell immunotherapy still has some challenges in the fight against solid tumors. It is mainly related to the immunosuppression of adoptive T cells in solid tumors. When T cells are inhibited, T cells show low targeting and low activation. It can lead to a sharp reduction of potent toxins secreted by T cells, such as perforin and granzyme B [98]. These toxins are mainly secreted by some natural killer cells and cytotoxic T lymphocytes (CTL cells) after activation [98]. When activated CD8+T cells bind specifically to tumor cells, and CD8+T cells release perforin and granzyme B [99] perforin and granzyme B can perforate the target cells, destroy the integrity of the cell membrane, and then kill the tumor cells. To find an effective strategy against solid tumors, Zhao et al. designed lysosome response nanoparticles (LYS-NP) loaded with anticancer proteins (Figure 5) [100]. LYS-NP can target adoptive T-cell lysosomes. T cells carrying anticancer proteins in lysosomes are known as adoptive T-cell vehicles (ATVs). The researchers used a metal-organic framework (ZIF-8) that can be degraded in an acidic environment as the core of the LYS-NP. Perforin and granzyme B were then loaded into LYS-NP. To deposit calcium ions on the surface or interior of ZIF-8. Calcium ions or calcium carbonate (CaCO_3_) would induce ZIF-8 mineralization, making it biocompatible and auto-degradable under an acidic environment. The degradation of mineralized ZIF-8 to produce Ca^2+^, Ca^2+^ enhances the function of perforin and granzyme B. Finally, the surface mineralized ZIF-8 was coupled with a lysosomal targeting aptamer (CD-63-APT) so that it could be specifically targeted to lysosomes. In this way, ZIF-8 can target cell lysosomes. In the acidic environment of lysosomes (pH = 5.0), LYS-NPs are degraded and release the perforin, granzyme B and Ca^2+^ stored in lysosomes. When ATVs reach the tumor site, they are activated by tumor cells, which in turn release perforin, Granzyme B, and calcium ions, eventually killing the tumor cells.

In recent decades, therapeutic vaccines have had particular advantages, including in tumor treatment [101,102]. An effective therapeutic vaccine would boost the immune system and eliminate target cells via CTLs [103,104]. Cellular immunity must be guided during this process. This requires antigen-presenting cells (APCs) to capture and further present disease-associated antigens to CD8 T cells via major histocompatibility complex (MHC) I molecules [105]. Unfortunately, the function that triggers adaptive immune response is an unsolved problem. Internalized exogenous antigens are often easily degraded in the lysosomes. This guides the MHC II pathway [106,107]. Finally, the activation of CD4 T cells results in a subsequent humoral response, rather than the desired cellular immunity. Therefore, a therapeutic vaccine that activates the MHC I pathway is needed. The MHC I pathway can efficiently deliver exogenous antigens and further cross-present them to APCs. With the rapid development of nanomedicine, Wang et al. proposed a simple one-pot method to prepare antigen (OVA)-doped CaCO3NPs (OVA@NP, =500 nm) for high-performance antigen delivery and cross-presentation [108]. OVA@NP acts as a nano-missile to efficiently transfer OVA into APCs. After reaching the acidic lysosome, OVA@NP with vaterite form and hierarchical structure significantly facilitates the decomposition reaction. The large amount of CO_2_ generated mechanically disrupts the lysosomal membrane. After disruption of the lysosomal membrane, OVA is released into the cytoplasm, enabling the cross-presentation of antigens. In addition to lysosomal escape, the resulting explosive force of CO_2_ also triggers autophagy. During this process, the remaining OVA in the lysosome is encapsulated by the autophagosome together with the cytoplasmic proteasome. This further enhances the effects of cross-presentation. The results showed that OVA@NPs achieved effectively enhanced cellular responses. OVA@NP promoted both CD8 T-cell proliferation and cytotoxic lysis. Ultimately, OVA@NP can improve tumor regression and prolong survival time. It provides a new way to develop safe and effective therapeutic vaccines to treat tumors.

## 5. Conclusions

We reviewed the research progress of nanoparticle responses to lysosomal function and environment in tumor therapy and analyzed their advantages and disadvantages. Based on lysosome function, researchers have designed various types of nanoparticles to destroy lysosomes, achieve lysosome escape and ultimately kill tumor cells. These nanoparticles were designed as nano-drugs with low toxicity or as nanocarriers that respond to endogenous stimuli (enzymes, pH, peptides and redox) or exogenous stimuli (temperature, light, ultrasound and magnetism). They can target the lysosomes of tumors or immune cells. Eventually, they kill tumor cells directly or indirectly. The discovery of lysosome-responsive functional nanoparticles provides a new direction for tumor treatment. It brings the treatment of tumors one step closer, from experimental to clinical.

Although nanotherapeutics that target lysosomal function have achieved much progress in tumor research, many problems remain. (1) Animal and cellular experiments cannot fully replicate tumor models owing to the complexity of the human body. Therefore, to better evaluate the biological activity, safety and effectiveness of nanoparticles in tumors, we suggest that data between different cell lines and species should be correlated. To create tumor models that are similar to human tumors, tissue-engineered constructs can be used as preclinical models to replicate the tissue tumor model. (2) Magnetically responsive metal nanoparticles can achieve the “drug-free” destruction of tumor lysosomes. However, the safety of metal nanoparticle accumulation in humans should be thoroughly evaluated. (3) Both patients and tumors are heterogeneous, but this heterogeneity was not stratified in the model replication studies.

Furthermore, the current knowledge and research in this direction may be just the tip of the iceberg. Lysosome functions are complex. Lysosomal membranes contain hundreds of integral and peripheral proteins. The role of these proteins in tumor cells is also unknown and many questions remain. For example: (1) what other nutrient and ion transporters, scaffolds and signalling proteins might associate lysosome function with tumor cells? (2) In addition to the responses to lysosomal function (lysosomal destruction and lysosomal escape) reviewed here, are there other pathways that mediate lysosomal function to play a role in the tumor microenvironment, tumorigenesis and metastasis? (3) Do lysosome-related organelles such as cytotoxic T-cell granules, melanosomes and platelet dense bodies play an important role in tumors? (4) Do different lysosomal subtypes have different functions in tumors?

## Figures and Tables

**Figure 1 cells-11-03348-f001:**
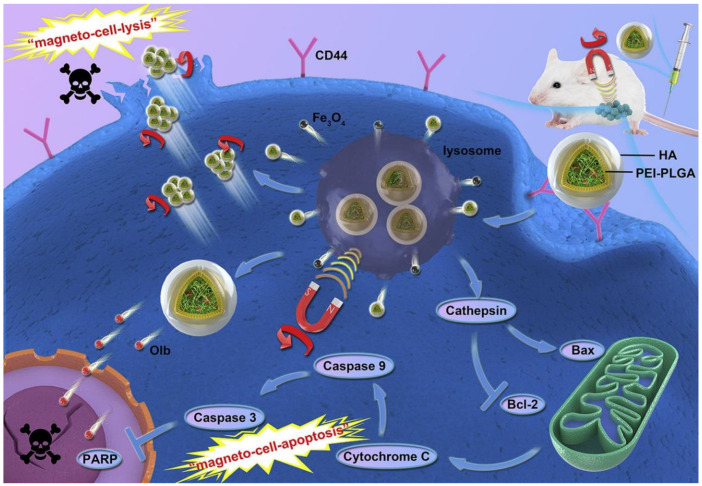
A multifunctional magnetic nanosystem based on the “two strikes” effect for synergistic anticancer therapy in triple-negative breast cancer [47].

**Figure 2 cells-11-03348-f002:**
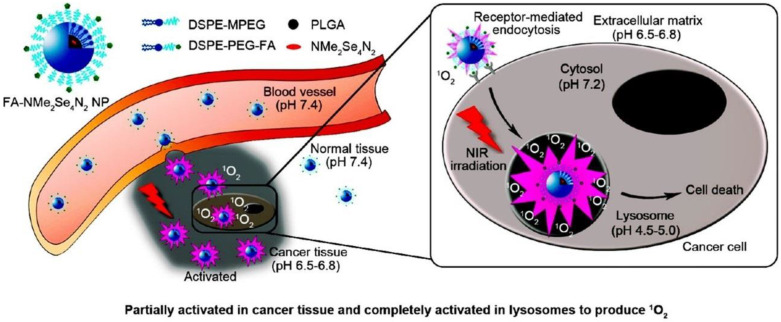
FA-NMe_2_Se_4_N_2_ NPs partially activated in cancer tissue and completely activated in lysosomes to produce ^1^O_2_ [54]. FA-NMe_2_Se_4_N_2_ NPs for highly selective Near-Infrared PDT against tumor.

**Figure 3 cells-11-03348-f003:**
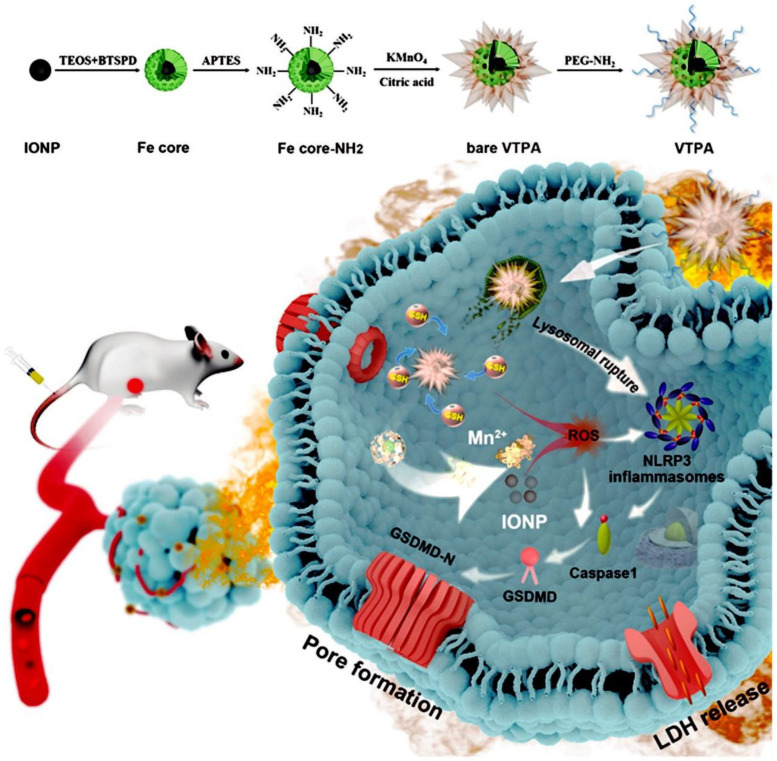
Schematic illustration of the design and therapeutic mechanism of virus-spike tumor-activatable pyroptotic agent (VTPA) [66].

**Figure 4 cells-11-03348-f004:**
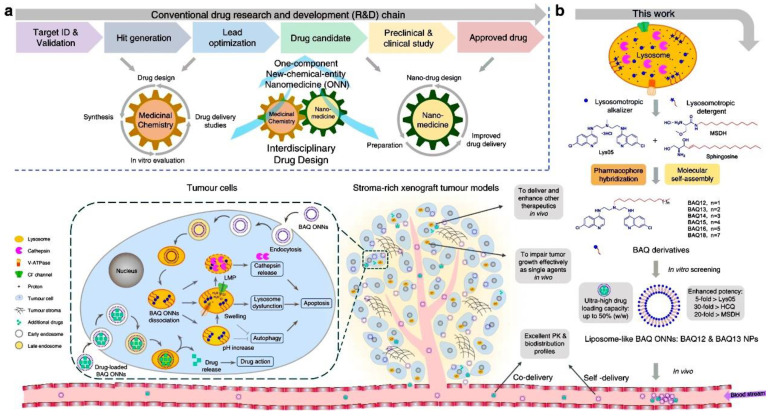
Schematic illustration of the proposed drug design strategy and the current study [80]. (**a**) Design of ONN. (**b**) The mechanism of self-assembled BAQ ONNs for tumor treatment.

**Figure 5 cells-11-03348-f005:**
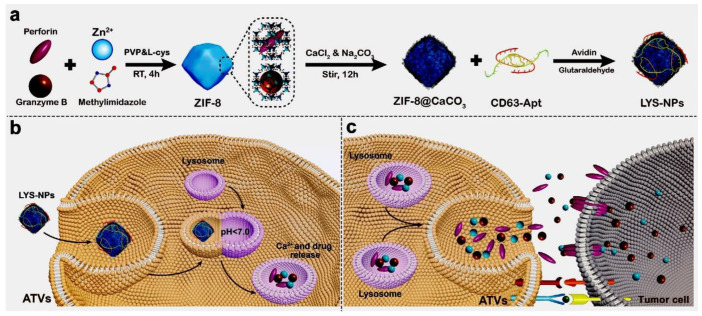
Schematic diagram of the design and synthesis of LYS-NPs and their function [100]. (**a**) Design and Synthesis of LYS-NPs. (**b**) Preparing ATVs with LYS-NPs. (**c**) Mechanism of ATVs killing tumor cells.

## Data Availability

Not applicable.

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
