# Peer review of "Nanomaterials Respond to Lysosomal Function for Tumor Treatment"

_cells, 2022, doi:10.3390/cells11213348_

Round 1
Reviewer 1 Report
This was an informative review paper highlighting the significance of lysosomes, nanomaterials, and the potential applications of nanomaterials and targeting lysosomes for cancer tumor treatment. The manuscript was generally well-written and presented the latest progress on nanoparticles and how they are designed to affect lysosome function in the context of specific cancer tumor types.
At the end of the review they made the statement that "Although nanotherapeutics that target lysosomal function have achieved much progress in tumor research, many problems remain. Animal and cellular experiments cannot fully replicate tumor models owing to the complexity of the human body." It would have been helpful if they added a thoughtful section detailing future directions or ideas on how the field should move forward, and the directions and ideas that may solve these complex problems. Adding such a section would increase the effectiveness of the paper and demonstrate the authors' knowledge in the field.
The paper needs to address some of the limitations of nanomaterials in vivo including the toxicity of nanomaterials themselves to normal cells as well as nanodrug delivery to older populations. Nanodrug delivery is a developing field but some of the current limitations need to be addressed.
One of the main biological challenges facing nanoparticles is controlling the passage of nanoparticles across biological barriers and into target cells limiting the effectiveness of all nanoparticle formulations. How might these nanomaterials discussed in this review improve target delivery effectiveness on a systemic level?
In the title and in the Abstract, there were some spacing issues where words ran together.
On lines 83-84 they state that nanocarriers are easily broken down. This was a surprising statement because basic drugs and basic nanomaterials get stuck in lysosomes and many substances appear to linger unmetabolized in lysosomes for extended periods of time. This also relates to the paragraph starting on line 183.
On line 125 they refer to AMF but they did not indicate to what this abbreviation referred.
It is not clear to me how they could present information on iron, reactive oxygen species and cell death and not include in their description of cell death mechanisms a section on ferroptosis.
Author Response
Please refer to the attachment, thank you!

Reviewer 2 Report
The review entitled “Nanomaterials respond to lysosomal function for tumour treatment” summarized the research progress on possibility to use nanoparticles to target lysosomal function and showed their advantages and disadvantages in antitumour therapy.
The review is clear and relevance to the field. There are not similar review published recently.
The references are recent publications and relevant. The number of selfe-citations is right.
The figures and the schemes clearly show the data and are easy to understand.
I have got a curiosity: How are the lysosomes of normal non-cancer cells modified following treatment with these nanomaterials?
After correcting some typos, the review can be accepted in its current form
Author Response

(The authors gave the same response as above.)

Round 2
Reviewer 1 Report
The responses and edits were adequate. I have no further issues.